# A Preliminary Study on Early Detection of Oral Cancer with Opportunistic Screening: Insights from Dental Surgeons in Sri Lanka

**DOI:** 10.3390/cancers15235511

**Published:** 2023-11-22

**Authors:** Dewasundara Wijenarayana Vishwa Nimanthi Dissanayaka, Konara Mudiyanselage Shashika Lakmali Wijeratne, Kodituwakku Arachchige Don Kaushal Devin Amarasinghe, Ruwan Duminda Jayasinghe, Primali Rukmal Jayasooriya, Balapuwaduge Ranjit Rigobert Nihal Mendis, Tommaso Lombardi

**Affiliations:** 1Teaching Hospital Karapitiya, Galle 80000, Sri Lanka; dwvnd@dental.pdn.ac.lk; 2Department of Oral Pathology, Faculty of Dental Sciences, University of Peradeniya, Peradeniya 20400, Sri Lanka; shashikmsl@gmail.com (K.M.S.L.W.); primalijaya@dental.pdn.ac.lk (P.R.J.); ranjitm@bluewin.ch (B.R.R.N.M.); 3Institution of Engineers, Wijerama Mawatha, Colombo 00700, Sri Lanka; amdevin@eng.pdn.ac.lk; 4Department of Oral Medicine and Periodontology, Faculty of Dental Sciences, University of Peradeniya, Peradeniya 20400, Sri Lanka; ruwanja@dental.pdn.ac.lk; 5Unit of Oral Medicine and Oral Maxillofacial Pathology, Division of Oral and Maxillofacial Surgery, Department of Surgery, University of Geneva & University Hospitals of Geneva, 1205 Geneva, Switzerland

**Keywords:** early detection, opportunistic screening, oral cancer, oral cancer prevention, oral potentially malignant disorders

## Abstract

**Simple Summary:**

In Sri Lanka, the authors are encouraging a cost-effective way to reduce the number of people developing oral cancer, which is a common cancer among men. The authors wanted to find out how well dentists can find signs of oral cancer and potential problems in their patients, and they did this by asking dentists questions in an online survey. Surprisingly, most of the dentists thought it would be better to only check people who are at a higher risk for oral cancer instead of checking everyone. Only about two thirds of the dentists regularly look for signs of oral cancer when people come for dental check-ups. The authors found that these dentists, on average, had noticed about 34 patients with concerning signs in the past year. Almost all the dentists believed that they needed more training to spot these problems. So, while opportunistic screening at the dentist’s office is good, dental professionals also need to focus on other ways to prevent oral cancer, like checking high-risk groups and giving dentists more confidence to spot potential issues.

**Abstract:**

In Sri Lanka, opportunistic screening is encouraged as a cost-effective tool to bring down the prevalence of oral cancer, which is the most common cancer among males. The objectives of the study were to determine the practices, attitudes, and level of competency of dental surgeons regarding the early detection of oral cancer and oral potentially malignant disorders (OPMDs) through opportunistic screening. A prospective study was conducted online via a Google form using a pretested, self-administered questionnaire of 22 close-ended questions and 3 open-ended questions. Out of the 137 dental surgeons who responded, 88% (121/137) of the participants believed that screening high-risk target groups would be more effective in the early detection of oral cancer rather than opportunistic screening. Only 64% (88/137) of the participants frequently check for oral cancer and OPMDs when patients visit for dental treatment. Participants recalled an average of 34 patients (4628/137) with clinically suspicious lesions being diagnosed during examination at general dental practice during the past year, and 98% (134/137) of the participants believed that they should receive additional training in order to identify and diagnose clinically suspicious OPMDs and oral cancer. Opportunistic screening in general dental practice as an oral-cancer prevention strategy is appreciable, but due emphasis should be given to other prevention strategies such as population screening and screening high-risk target groups. The level of confidence of general dental practitioners in the early detection of oral cancer has to be raised in order to achieve higher standards in oral cancer prevention through opportunistic screening.

## 1. Introduction 

Oral and oropharyngeal cancers are a significant, growing health problem, ranked the sixth most common cancer globally [1,2,3,4]. However, particularly in South Asian countries, oral cancer is the most common cancer among men which contributes to about 25% of all new cases of cancer detected each year [1]. In addition, the mortality rate for oral cancer is appreciably high with a 5-year survival rate of only 50% [1,2,5]. Oral squamous cell carcinoma (OSCC) accounts for 95% of cancers occurring in the oral cavity. The most important etiological factors for OSCC are betel quid chewing, smoking and excess consumption of alcohol [2], betel quid chewing, and smoking [2]. The contribution of only primary prevention is accounted to be largely ineffectual as evidenced by the increased use of tobacco, alcohol, and areca nut despite educational and public awareness programmes [1,6,7]. 

Although the direct accessibility and visibility of the oral cavity for examination greatly facilitate the early detection of OSCC and oral potentially malignant disorders (OPMDs), in Sri Lanka, a large majority, or approximately 60%, of OSCCs are diagnosed in advanced clinical stages [3,5,8]. The delay of patients in seeking professional advice is linked with the late presentation of OSCC leading to poor prognosis and decreased survival rates due to loco-regional and distant metastasis [1,8]. The proportion of patients who present with advanced disease stages has not changed during the past 40 years, and most studies have shown no improvement in survival rates even though a considerable proportion of the population receives regular oral examination and routine dental care [1,9]. 

Screening as a secondary prevention protocol is worthy of consideration because it is a well-established fact that the early diagnosis of OSCC can result in higher survival rates and better prognosis. Opportunistic oral cancer screening can be defined as a systematic inspection of the oral cavity and head and neck region to identify clinical signs of OPMD and OSCC during the routine oral examination of dental patients who present without symptoms of disease [8,10,11]. Opportunistic screening has many advantages over other types of screening because it is largely cost-effective and easy. However, the adequacy of mere opportunistic screening as an oral cancer prevention strategy in the considerably socio-economically deprived population in Sri Lanka is uncertain. Furthermore, it is noteworthy that OSCC has the potential to be preceded by a clinically detectable potentially malignant stage that often manifests as an asymptomatic lesion in the early part of its natural history. Thus, accurate diagnosis and timely treatment may prevent the malignant transformation of such lesions [10,12,13]. Therefore, the possibility of oral cancer/OPMDs being detected early and treated in early stages through opportunistic screening leading to a good prognosis and acceptable quality of life remains high.

Understanding the attitudes, practices, and level of competency of dental surgeons about opportunistic screening for OSCC and OPMDs is essential for assessing the achievability of the early detection and prevention of oral cancer with opportunistic screening. Furthermore, the identification of requirements for dental surgeons is paramount to the formulation of new protocols such as training programmes to improve the quality and quantity of opportunistic screening.

Sri Lanka is a developing nation in which a substantial proportion of the population does not seek regular dental treatment or routine oral examination [14]. Hence, the prevalence of oral cancer and OPMDs detected in general dental practice attendees remains unknown, and the attributes of the prevalence of oral cancer in the population are also questionable. The main objective of this study is to assess the practices, attitudes, and competency levels of dental surgeons in the early detection of OSCC and OPMDs through opportunistic screening. The specific objectives of this study are to determine the attitudes of general dental practitioners toward risk behaviour assessment and tobacco cessation education and assess the competency levels of dental surgeons in diagnosing OPMDs/OSCC. The adequacy of opportunistic screening needs to be emphasised qualitatively and quantitatively to weigh its impact as an appreciable method for the secondary prevention of oral cancer. This study potentiates further intervention and explores whether opportunistic screening is a realistic alternative to population screening and screening high-risk target groups.

## 2. Materials and Methods

Dental surgeons in Sri Lanka who practice dentistry in government and private settings with access to the World Wide Web were the participants of this study. Dental surgeons who refrain from practicing dentistry in Sri Lanka and who do not have access to the World Wide Web were excluded from this study. A self-administered online questionnaire modified minimally for readability and layout was created with Google Forms. The questionnaire was administered via the World Wide Web. The survey link was disseminated as a Uniform Resource Locator (URL) or web address among dental surgeons via social media, email, online messaging mobile application groups, etc. Google Forms is a web-based survey tool which is provided by Google Inc. that allows users to create surveys, questionnaires, and forms for various purposes. The forms were designed and shared with participants, and responses were submitted electronically. The Google form was distributed multiple times among dental surgeons; however, one participant was restricted to providing only a single response.

A cover letter was included with the questionnaire, which described the study in detail and the members of the work team and their contact details. Instructions were given for completing the questionnaire, and the confidentiality and anonymity of the data provided were assured. The questionnaire was first pre-tested among a convenient sample to ensure the clarity of interpretation and ease of completion by the participants. No compensation was provided to the participants for their involvement in this study.

The questionnaire used in this study consisted of 22 close-ended questions and 3 open-ended questions. The questionnaire was designed in 5 parts which include the demographic data of participants, the attitudes of participants regarding the risk behaviour assessment and tobacco cessation of patients, the diagnosis/examination of OPMD and OSCC, the level of competency of participants, and the requirements for the participants to improve opportunistic screening. The questionnaire was outlined according to the objectives of the study. The four stages of competence, or the “conscious competence” learning model [15,16], were used to assess the level of competency of dental surgeons, and a Likert scale was used to assess the attitudes of participants [17]. The data management and statistical analysis were performed using the statistical software SPSS version 19.0. Frequencies and percentages were obtained for categorical data. The relationship between number of patients examined and number of patients detected with OPMD/OSCC was analysed using Pearson’s correlation coefficient. Ethical clearance for this study was obtained from the Ethics Review Committee of the Faculty of Dental Sciences, University of Peradeniya (ERC/FDS/UOP/1/2020/15). 

## 3. Results

One hundred and thirty-seven dental surgeons employed in 22 different districts and within the age range of 25–68 years completely responded to the questionnaire administered online. Participants who did not complete their questionnaires and those who did not fill out the questionnaires accurately were excluded from the study. To summarize, 37% (51/137), 24% (34/137), and 18% (25/137) of the participants had 5–10 years, 11–20 years, and <5 years of working experience, respectively. Only 7% (7/137) of the participants had working experience of more than 30 years. Moreover, 44% (60/137) were government service employees, whereas 16% (22/137) of the participants were full-time private practitioners, and 36% (50/137) of the participants were engaging in both private practice and government service (Table 1).

In the context of the examination and diagnosis of OPMDs/oral cancer by the participants, the average number of patients seen by a dental surgeon per week in government service and private practice was 119 (11,350/95; range 10–600) and 56 (4318/77; range 4–250), respectively. Only 64% (88/137) of the participants frequently check for oral cancer when patients visit for dental treatment. Participants recalled an average of 34 (4628/137) patients with clinically suspicious lesions diagnosed during an examination at a general dental practice during the past year. Buccal mucosa (41%; 61/137) followed by the tongue (22%:31/137) were the most common sites where participants examined most often (Figure 1). Oral submucous fibrosis (64%; 89/137) followed by leukoplakia (21%; 29/137) was the most common clinical presentation identified (Figure 2).

To determine whether there is a relationship between the number of patients seen at regular dental practice and the number of patients with detected OPMDs/oral cancer, the correlation coefficient was used. The dataset used in this study comprised observations of 137 participants from 22 districts in Sri Lanka. The variable X represents the total number of patients seen per dental surgeon per year, while variable Y represents the total number of patients with detected OPMDs per dental surgeon per year. Pearson’s correlation coefficient was used to assess the strength and direction of the association between these two variables. The computed Pearson’s correlation coefficient between X and Y was found to be *r* = 0.428, indicating a weak negative relationship between the two variables. The scatter plot does not depict a linear trend, and the line is not to be used for prediction (Figure 3). To assess the statistical significance of the correlation coefficient, we conducted a two-tailed *t*-test. The resulting *p*-value was 0.047, which is less than 0.05, indicating that the correlation between X and Y is statistically significant at the 5% significance level. The weak negative correlation (*r* = 0.428) observed between X and Y suggests that changes in X are not associated with corresponding changes in Y.

After analysing responses from participants in 22 out of Sri Lanka’s 25 districts, it is evident that while most patients tend to seek regular dental treatment in districts like Colombo, Kandy, and Gampaha which are urban and suburban communities, a higher incidence of patients with OPMDs was observed in districts such as Mannar, Nuwara Eliya, and Polonnaruwa, which are generally rural communities (Table 2, Figure 4).

Regarding the attitudes and practices of the participants regarding risk behaviour assessments and tobacco cessation advice, 73% (100/137) of the participants inquired about betel chewing frequently while examining the patients. Only 46% (63/137) of the participants inquired about smokeless tobacco use. While 52% (71/137) of the participants inquired about areca nut use specifically when betel chewing was mentioned, 15% (21/137) of the participants never inquired about areca nut use. Only 23% (32/137) of the participants inquired about alcohol use, and 33% (45/137) of the participants never inquired about alcohol use. Only 69% (95/137) of the participants frequently provided tobacco cessation advice at general dental practice. 

According to the conscious competence scale, 8% (11/137) of the participants did not understand how to give tobacco cessation advice and did not believe it was a necessity. However, 64% (88/137) of the participants had the understanding to deliver tobacco cessation advice but requested more training. Additionally, 37% (51/137) and 27% (37/137) of the participants believed that patients’ lack of interest and lack of time were the most likely reasons not to provide adequate tobacco cessation advice in general dental settings, respectively. Only 58% (79/137) of the participants agreed that opportunistic screening is achievable in the general dental setting. Further, 88% (121/137) of the participants believed that population screening and screening target groups will be more effective in the early detection of oral cancer rather than opportunistic screening, and 95% (130/137) of the participants agreed that the patients at the highest risk for oral cancer are the least likely to be regular attendees of a general dental practice. 

Considering the level of competency of the participants, 47% (64/137) of the participants found it uncomfortable to perform a biopsy in general practice, 58% (79/137) of the participants had not attended any educational programme on OPMD/oral cancer within last 5 years, and 72% (99/137) of the participants had a satisfactory level of confidence in identifying oral cancer presentation. According to the conscious competence scale, 59% (81/137) of the participants believed that they understood how to examine for cervical lymphadenopathy but requested more practice, while 6% (8/137) of the participants seemed to think it is not necessary. Further, 98% (134/137) of the participants believed that they should be trained more to identify and diagnose clinically suspicious OPMD/oral cancer lesions in general dental practice (Figure 5).

## 4. Discussion

Oral cancer is a major public health burden. Following its recognition as the 16th most common malignant neoplasm in the world, there is an increasing trend of oral cancer affecting young men and women [18]. Public awareness regarding OSCC and OPMDs is poor, and many patients present with late-stage disease, contributing to high mortality. Oral cancer is often preceded by a clinical premalignant phase that may be observed visually, thus providing the opportunity for early detection, which reduces morbidity and mortality. Oral cancer is linked to social and economic status and deprivation, with the highest rates occurring in the most disadvantaged sections of the population [18,19]. An analysis of outcomes of patients with early-stage oral cancer indicates that these patients have a good prognosis [20,21] and improved rates of survival and quality of life. However, early-stage oral cancers and OPMDs are often asymptomatic; thus, there is a reduced likelihood for the public to seek care, and, therefore, screening provides an opportunity for early detection.

Our research aims to explore the intricate relationship between the number of patients seen by dental practitioners and the detection of potentially malignant oral disorders (OPMDs) and oral cancer in government and private settings in Sri Lanka. This investigation encompassed 137 participants representing a diverse range of dental surgeons practicing across the 22 districts in Sri Lanka. The computed Pearson’s correlation coefficient (*r* = 0.428) indicated a weak negative relationship between the total number of patients seen per dental surgeon per year (variable X) and the number of patients with detected OPMDs/oral cancer per dental surgeon per year (variable Y). This suggests that the number of patients seen by dental surgeons in the dental setting is not strongly associated with a corresponding number of patients with detected OPMDs/oral cancer at the general dental practice with opportunistic screening. Additionally, it is worth emphasising whether this finding reflects the fact that the patients at the highest risk for oral cancer/OPMDs are the least likely to be attendees of dental practice in a developing country with low socio-economic growth such as Sri Lanka. This finding alone warrants the consideration of screening high-risk groups in the population and the necessity of organised cancer screening programmes such as population-based screening (both through home visits or by invitation to attend screening events). Integrating oral cancer screening with general health screenings and screening at the place of work (e.g., estate workers) can be considered. In comparison to the general-population-based screening used in most research, risk-based modelling for screening “at-risk” populations offers greater efficacy [22].

Following a study on “opportunistic” screening conducted in Cuba between 1982 and 1990, over 10 million people were examined, of whom 0.3% were “screen positive” [22]. In a randomised controlled trial conducted in Kerala from 1994 to 2009 targeting a high-risk population, after four rounds of screening, the authors reported a sustained reduction in mortality of 81% and a reduction in the incidence of oral cancer in the screened population compared with a control population [23]. Additionally, 2 million Taiwanese adults who were smokers and/or betel quid chewers were invited for oral examination by a dental surgeon or primary health care physician twice a year. As a result, 55% attended the screening, and 4110 were confirmed to have oral cancer at their first screening [24].

Opportunistic screening for oral cancer/OPMDs is largely performed at dental clinic setups but not in other primary health care settings in Sri Lanka. The workforce of dental surgeons available warrants additional training. Appropriate training should be provided to increase the accuracy of diagnosis of oral cancer/OPMD with clinical presentation because a considerable majority of dental surgeons demand further training. Strengthening undergraduate curricula of medical, dental, nursing, and allied health care training programmes on oral cancer prevalence, presentation, detection, and prevention is mandatory. In high-income countries like the United States, the role of the dental team in opportunistic screening cannot be underestimated [25,26,27]. However, in developing countries, there is a poor benefit to the population with a lack of access to care and those who attend primary dental care clinics irregularly [28].

In accordance with a study conducted by Conway et al., a pronounced and statistically significant association was identified between low socioeconomic status and the risk of oral cancer compared to lifestyle risk factors [29]. The results of the study reflect that a substantial number of dental surgeons find screening asymptomatic individuals using systematic visual oral examinations to detect oral cancer and OPMDs feasible in general dental practice. However, the recalled number of patients with the disease was exceptionally low. Whether this can be attributed to the socio-economic status of the country is highly disputable, as reflected in the results obtained from the participants from different districts in Sri Lanka.

Regarding the attitudes and practices of the participants, we observed variations in the frequency of risk behaviour assessment and tobacco cessation advice provision. While a significant proportion of participants inquired about betel chewing during patient examinations, a lower percentage inquired about smokeless tobacco use. These variations highlight the need for standardised protocols and continued education in this regard. It is noteworthy that a considerable percentage of participants expressed discomfort with performing biopsies in general practice and had not attended educational programmes on OPMD/oral cancer in the last 5 years. This underscores the importance of continuous professional development and training to enhance the diagnostic skills of dental surgeons.

Intensive smoking-cessation intervention in the dental setting is proven to be effective [30]. In this study, the challenges of providing tobacco cessation advice were also identified. While many participants believed in its importance, some felt uncomfortable giving such advice, suggesting the need for training in this critical aspect of preventive care. Finally, our study emphasises the perspectives of the participants on screening strategies. Most participants believed that population screening and targeting high-risk groups would be more effective in the early detection of oral cancer compared to opportunistic screening. This insight could inform future public health initiatives. Regional disparities and challenges in risk behaviour assessment and tobacco cessation advice provision require further attention. Continuous training and standardised protocols should be considered to improve the competency of dental surgeons and promote early detection and prevention efforts in oral health care.

Despite the strengths of this study, which elaborated the achievability of opportunistic screening in the setting of general dental practice for the early detection of OSCC and OPMD using a considerable study population of dental surgeons in Sri Lanka with due emphasis on the necessity of implementing alternative screening strategies such as screening targeted and high-risk groups, there were several limitations of this study associated with difficulties in the comprehension and interpretation of the questionnaire and with accuracy and reliability because of the approximate number of patients recalled.

Whether this study adequately captures regional variations in healthcare infrastructure, patient demographics, and socioeconomic factors is questionable because this study relied on voluntary participation. Further studies to evaluate the effectiveness and feasibility of different screening strategies, including opportunistic screening, population-based screening, and targeted high-risk group screening, can be suggested. Moreover, the scientific community can deepen its understanding of oral cancer screening practices, identify areas for improvement, and contribute to the development of effective strategies for early detection and prevention through comparative analysis with other health care professionals, evaluating screening strategies, and exploring global health disparities.

## 5. Conclusions

The consideration of opportunistic screening as an alternative to population screening and screening high-risk target groups is highly questionable in a developing country like Sri Lanka because the population at high risk for oral cancer is less likely to comprise regular attendees of general dental practice. The achievability and adequacy of opportunistic screening must be further emphasised qualitatively and quantitatively. The level of confidence of general dental practitioners in the early detection of oral cancer must be improved with further training to achieve higher standards in oral cancer prevention through opportunistic screening. Further research and interventions in this area are warranted to improve oral cancer prevention and early detection in the country.

## Figures and Tables

**Figure 1 cancers-15-05511-f001:**
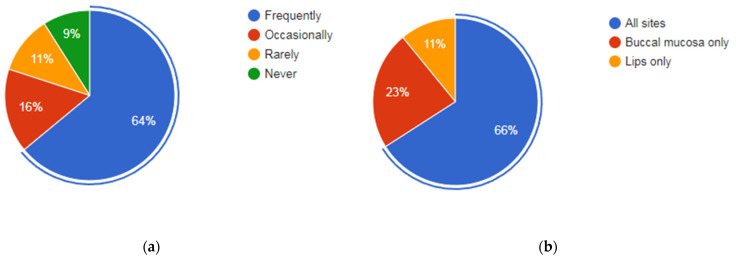
(**a**) Frequency of participants checking for oral potentially malignant disorders/oral cancer when patients visit for routine dental treatment. (**b**) Sites where participants pay most attention during routine dental treatment.

**Figure 2 cancers-15-05511-f002:**
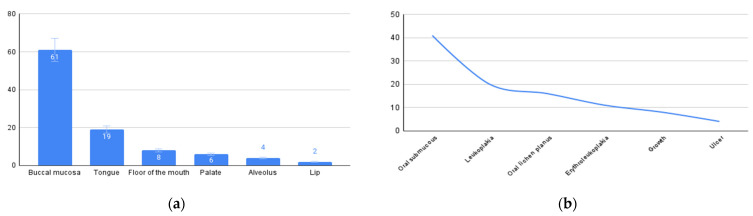
(**a**) Buccal mucosa is the most common site in the oral cavity to be examined during routine dental treatments. (**b**) Oral submucous fibrosis is the most common presentation identified when OPMDs/oral cancers were detected during routine dental treatment.

**Figure 3 cancers-15-05511-f003:**
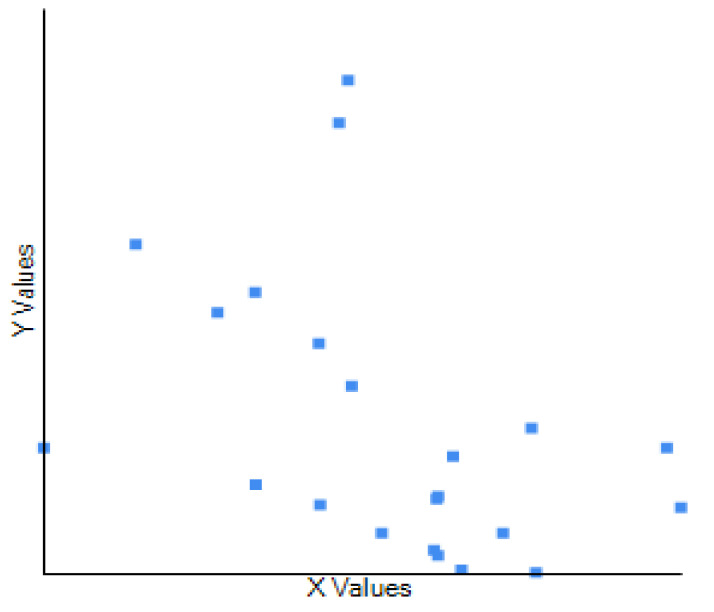
Negative and weak relationship between the total number of patients seen at general dental practice and the number of patients detected with clinically suspicious lesions with opportunistic screening. The scatter plot does not show a linear trend, the line should not be used for prediction.

**Figure 4 cancers-15-05511-f004:**
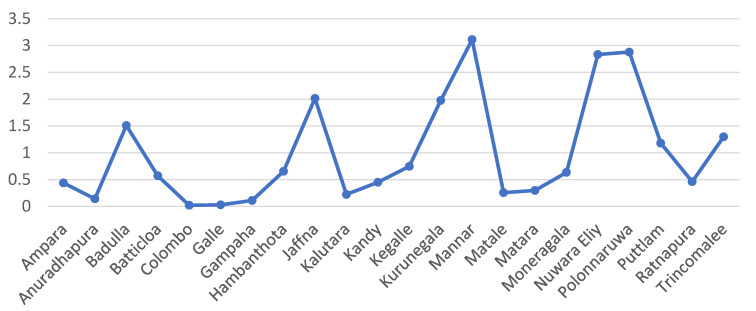
As recalled by participants, even though most patients seek regular dental treatment in districts such as Colombo, Kandy, and Gampaha, increased numbers of patients with OPMDs were detected in districts such as Mannar, Nuwara Eliya, and Polonnaruwa.

**Figure 5 cancers-15-05511-f005:**
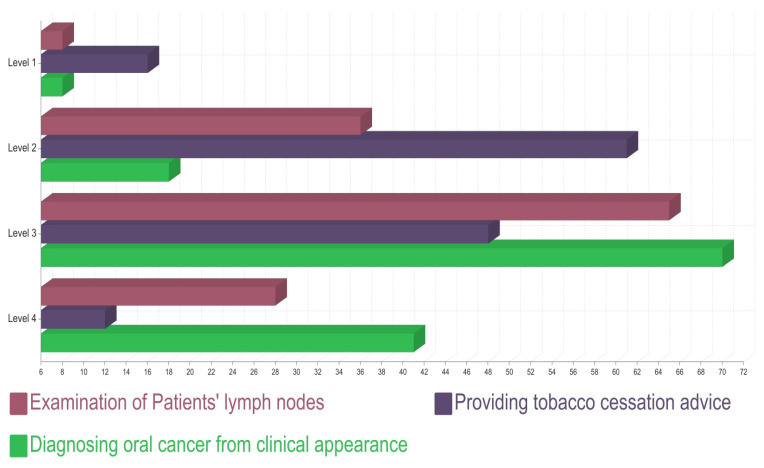
Levels of competency of participants according to conscious competence scale; Level 1: UNCONSCIOUS INCOMPETENCE (I don’t understand how to do it and I don’t think it is necessary), Level 2: CONSCIOUS INCOMPETENCE (I don’t understand how to do it but I think it is necessary), Level 3: CONSCIOUS COMPETENCE (I understand how to do it, but I need more practice), Level 4: CONSCIOUS COMPETENCE (I have a lot of practice doing it and I can even teach it to others).

**Table 1 cancers-15-05511-t001:** Demographic details of participants.

Age	Males (n)	% of Participants	Females (n)	% of Participants	Total Number of Participants (N)
26–30	24	28.24	12	23.08	36
31–35	25	29.41	16	30.77	41
36–40	17	48.57	11	21.15	28
41–50	11	18.03	9	17.31	20
51–60	6	17.65	4	7.69	10
61–65	2	28.57	0	0.00	2
	85		52		137
Duration of experience					
1–5 years	35	25.55			
6–10 years	61	44.53			
11–20 years	34	34.69			
>20 years	7	41.18			
	137				
Professional qualifications					
BDS only	98				
Postgraduate degree	17	12.41			
Postgraduate diploma	22	36.67			
	137				
Employment status of participants					
Government service only	60	46.58			
Private practice only	22	13.66			
Government service and private practice	55	39.75			
	137				

BDS: Bachelor in Dental Surgery, Postgraduate degree: participants who have followed a postgraduate degree programme in speciality related to dentistry following basic degree in Dentistry, Postgraduate diploma: participants who have followed a diploma course in Hospital Dental Practice or any other related to dentistry following basic degree in Dentistry.

**Table 2 cancers-15-05511-t002:** Study participants were 137 dental surgeons from 22 districts in Sri Lanka. Approximate number of patients seen at general dental practice and the number of patients with detected OPMDs.

District	Number of Dental Surgeons	App. Number of Patients Seen at General Dental Practice/Year/Dental Surgeon (X-Axis)	App. Number of OPMDs Detected/Dental Surgeon/Year (Y-Axis)
Ampara	3	6253	27
Anuradhapura	4	6234	9
Badulla	3	5417	82
Batticaloa	4	7886	45
Colombo	24	6959	1
Galle	8	6429	2
Gampaha	13	6263	7
Hambanthota	6	6370	42
Jaffna	3	4965	100
Kalutara	4	6723	15
Kandy	16	6263	28
Kegalle	6	6927	52
Kurunegala	7	4699	93
Mannar	2	5625	175
Matale	4	5864	15
Matara	8	7987	24
Moneragala	4	4970	32
Nuwara Eliya	3	4119	117
Polonnaruwa	2	5560	160
Puttlam	3	5651	67
Ratnapura	6	5427	25
Trincomalee	4	3466	45

## Data Availability

The data presented in this study are available on request from the principal investigator. The data are not publicly available due to ethical and privacy considerations.

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
