# Peer review of "A Preliminary Study on Early Detection of Oral Cancer with Opportunistic Screening: Insights from Dental Surgeons in Sri Lanka"

_cancers, 2023, doi:10.3390/cancers15235511_

Round 1

Reviewer 1 Report

Comments and Suggestions for Authors

I would like to congratulate the authors for conducting the present study which addresses a very relevant topic which is the oral cancer. Here goes a few concerns and comments to the authors:

I would recommend the authors to increase the number of keywords to 5, and place them in alphabetic order.

A proper study aims and objectives are missing from the end of the Introduction sentence.

How has the questionnaire delivered to the participants?

How was the questionnaire built? Who decided for which questions to make, and why were does questions made?

How many times were the questionnaires sent?

How did the authors manage the possible double participation (questionnaires repetition) from some participants?

Was there any compensation for participating in the questionnaire?

Table 1 needs a footnote legend for BDS abbreviation

Table 2 needs a footnote legend for OPMD abbreviation

In the last sentence of the Discussion the authors mention the study strengths. May the authors elaborate regarding which strengths they are talking about?

May the authors debate the generalization of the results?

May the authors debate about further studies outcomes in the end of the Discussion?

The final reference list is not according to the journal guidelines

Author Response

5 keywords were included in the list of keywords according to the alphabetical order.

  1. The study aims and objectives were added to the end of the Introduction section
  2. The method of delivery of questionnaires to participants is included in the first paragraph of Materials and Methods section.
  3. The basis of the questionnaire used in the study was included.

5 &6. The way the questionnaire was distributed among dental surgeons and how double participation was avoided was mentioned.

  1. The fact that any compensation was not provided for participants was mentioned.
  2. Footnote legend for BDS abbreviation was added.
  3. Footnote legend for OPMD abbreviation was added.
  4. The strengths of the study were emphasized in the latter part of the Discussion section.
  5. Generalization of the results highlighting potential challenges, limitations and recommendations was included in the final section of the Discussion section.
  6. Further studies to pursue in the same field of study were suggested in the final paragraph.
  7. The reference list was corrected according to the journal guidelines. Thank you.

Reviewer 2 Report

Comments and Suggestions for Authors

Methodology should be elaborated

Author Response

Materials and methods section was elaborated as suggested . Thank you..

Reviewer 3 Report

Comments and Suggestions for Authors

The study is well organized and presented is a good scientific manner. However, there are some modifications required for further consideration. 

- Please define the "Postgraduate degree" and "Postgraduate diploma" criteria. It is not clear from the terminology. 

- I do not think that figure 3 is necessary and the authors can add the information in the text. 

- Please add the future directions of the main study following this preliminary study.

Author Response

- ‘Postgraduate degree’ and ‘postgraduate diploma’ criteria were defined.

- Authors find the scatter plot (Figure 3) which depicts the weak correlation between the two variables important to be mentioned in the manuscript.

- Future directions and recommendations were included. Thank you

Round 2

Reviewer 1 Report

Comments and Suggestions for Authors

Dear authors, I have no further comment.